# Child Fluoride Varnish Programs Implementation: A Consensus Workshop and Actions to Increase Scale-Up in Australia

**DOI:** 10.3390/healthcare9081029

**Published:** 2021-08-11

**Authors:** John Skinner, Yvonne Dimitropoulos, Woosung Sohn, Alexander Holden, Boe Rambaldini, Heiko Spallek, Rahila Ummer-Christian, Stuart Marshall, Kate Raymond, Tom Calma AO, Kylie Gwynne

**Affiliations:** 1Poche Centre for Indigenous Health, The University of Sydney, Room 224 Edward Ford Building, Sydney 2006, Australia; yvonne.dimitropoulos@sydney.edu.au (Y.D.); boe.rambaldini@sydney.edu.au (B.R.); tom.calma@sydney.edu.au (T.C.A.); kylie.gwynne@sydney.edu.au (K.G.); 2The University of Sydney School of Dentistry, 2 Chalmers St, Surry Hills 2010, Australia; woosung.sohn@sydney.edu.au (W.S.); alexander.holden@sydney.edu.au (A.H.); heiko.spallek@sydney.edu.au (H.S.); 3Fluoride Varnish Initiative, Loddon Mallee Aboriginal Reference Group, Bendigo 3550, Australia; rahila.christian@bdac.com.au; 4Statewide Dental Services, SA Dental, Level 5 Roma Mitchell House, 136 North Tce, Adelaide 5000, Australia; stuart.marshall@sa.gov.au; 5Department of Health, Northern Territory Government, Level 7 Manunda Place, 38 Cavenagh Street, Darwin 0800, Australia; kate.raymond@nt.gov.au; 6Faculty of Medicine, Health and Human Sciences, Macquarie University, Sydney 2113, Australia

**Keywords:** fluoride varnish, dental assistants, Aboriginal and Torres Strait Islander, oral health, implementation science, scale-up

## Abstract

This paper presents the findings of the National Fluoride Varnish Workshop in 2018 along with subsequent actions to scale-up the use of fluoride varnish nationally in Australia. The use of fluoride varnish programs to prevent dental caries in high-risk child populations is an evidence-based population health approach used internationally. Such programs have not been implemented at scale nationally in Australia. A National Fluoride Varnish Consensus Workshop was held in Sydney in November 2018 with an aim of sharing the current work in this area being undertaken by various Australian jurisdictions and seeking consensus on key actions to improve the scale-up nationally. Forty-four people attended the Workshop with oral health representatives from all Australian state and territory health departments, as well as the Australian Dental Association (ADA) at both NSW branch and Federal levels. There was strong support for further scale-up of fluoride varnish programs nationally and to see the wider use of having non-dental professionals apply the varnish. This case study identifies key actions required to ensure scale-up of systematic fluoride varnish programs as part of a strategic population oral health approach to preventing dental caries among high-risk children who may not routinely access dental care.

## 1. Introduction

The purpose of this study is to outline progress towards the scaling up of fluoride varnish programs in Australia to prevent dental disease, particularly among high-risk children. The National Oral Health Plan acknowledges the importance of the wider adoption of fluoride varnish programs nationally as part of oral health promotion efforts for preventing dental disease, and the Plan sees these programs as complementary to the population health benefits of water fluoridation programs and regular dental visits [1]. The most recent national data report that, in 2017, approximately 89% of the Australian population had access to optimally fluoridated water [2]. A recent economic evaluation review of water fluoridation found that it had a caries reduction effect of 25 to 40% and was cost effective [3]. The increased use of fluoride varnish is also supported by state oral health plans, evidence-based oral health promotion and fluoride guidelines and dental and medical professional groups [1,4,5,6,7,8,9,10]. A review undertaken by the Cochrane Collaboration also supports the regular use of fluoride varnish to prevent dental caries [11].

Aboriginal and Torres Strait Islander children aged 6–14 years have on average 1.3 decayed, missing or filled tooth surfaces (DMFS) compared to 0.7 for non-Aboriginal and Torres Strait Islander children [12]. Much of this excess dental disease is linked to the long-term impact of colonisation, racism and historical trauma [13]. The social and emotional determinants of health are also reflected in generally lower levels of protective oral health behaviours, including toothbrushing and regular dental visiting. Aboriginal and Torres Strait Islander children and teens are also over-represented in those requiring removal or restoration of teeth under general anaesthetic. In 2016/17, the rate of potentially preventable hospitalisations due to dental conditions was highest in children aged 5–9 years, at 13.3 per 1000 population for Aboriginal and Torres Strait Islander children versus 9.3 per 1000 for non-Aboriginal children [14]. This high disease burden means that fluoride varnish is an important preventive strategy for these children even if they have access to the benefits of water fluoridation.

The most common fluoride varnish product for the prevention of dental caries in Australia is Duraphat^TM^, which contains 22.6 mg/mL fluoride in suspension [7]. The fluoride varnish is painted on all the surfaces of the teeth and “…forms a waxy film that adheres to the teeth until it is worn off by chewing or brushing” [7]. Fluoride varnish programs are usually pre-school or school-based programs where parents and guardians consent to their child having the varnish applied [5,7]. The process is very quick, non-invasive and painless, and large numbers of children can be seen in one day. While there are other fluoride varnish products on the market today, the majority are registered for treating dentine hypersensitivity rather than the prevention of caries [7].

The National Oral Health Plan recognises that the application of fluoride varnish is a safe, effective and efficient oral health promotion strategy for reducing dental decay in vulnerable populations [1]. The Plan includes references to fluoride varnish and/or its application by dental assistants under three foundation areas: oral health promotion, systems alignment and integration, and workforce development. In the oral health promotion section of the Plan, it is noted that communities without access to fluoridated water need access “…to fluoride in its other forms, including fluoride varnish programs and affordable oral hygiene products such as toothpaste” [1]. There is also good evidence that fluoride varnish programs complement the fluoride available in toothpaste and water fluoridation in high-risk groups [15,16]. Even with access to fluoride from these complementary sources, the risk of dental fluorosis and other side effects from fluoride varnish is negligible [17,18].

In supporting the use of fluoride varnish programs as a key oral health promotion strategy, the National Oral Health Plan acknowledges that additional workforce may be needed to support these programs, particularly as these projects are expanded and developed. The Plan notes the need to have a more flexible approach to utilising the skills of both the registered and non-registered dental workforce [1]. The National Oral Health Plan also notes that the “legislation covering the use of fluoride varnish and other products varies between jurisdictions, impacting on the ability to implement consistent programs that maximise access to an effective intervention. Relevant legislation and regulations including Poisons Acts, Health Acts and Radiation Safety Acts should be reviewed to remove unnecessary barriers to the provision of dental care” [1]. These barriers largely relate to the listing of fluoride varnish and related restrictions on who can obtain, possess and apply it with most jurisdictions limiting this to medical, dental and oral health practitioners.

### Objective

The aim of this study is to describe the current efforts to implement child fluoride varnish programs as discussed at a National Consensus Workshop, along with progress to date in addressing barriers to increased scale-up of programs, particularly for Aboriginal and Torres Strait Islander children.

## 2. Materials and Methods

Twenty four of the 44 attendees completed a pre-Workshop survey in SurveyMonkey [19], which allowed them to choose which working group they would like to be a part of along with providing details of current fluoride varnish projects and research. A literature review along with a review of the key health promotion actions of the National Oral Health Plan related to fluoride varnish yielded three key themes that were discussed at the Workshop. The three key themes explored by the workshop included:Workforce issues, including greater use of dental assistants and training courses;Prescription, eligibility, settings and protocol issues; andResearch and evaluation.

The key aims of the workshop were:To share information between organisations and jurisdictions with efforts to expand the use of fluoride varnish amongst school children in Australia;To develop greater consistency and scalability of fluoride varnish programs for Australian school children including protocols and criteria to target schools;To promote greater use of dental assistants to apply fluoride varnish, and the training of Aboriginal dental assistants in communities with a high Aboriginal child population;Work towards consistent legislation nationally that would allow suitably trained dental assistants and Aboriginal health workers to routinely apply fluoride varnish.

The 44 Workshop attendees were allocated to one of three working groups to discussion these themes in detail. Each group had a designated leader and reported back their findings to the wider group. These were summarised on the day and presented back to the whole group in the afternoon session. These findings form the basis of the tables presented in this case study.

### Ethics Approval

The Improving Child Oral Health: Scale-up and Standardisation of Fluoride Varnish Programs in Australia workshop and publication of the national consensus workshop outcomes were approved by the NSW Aboriginal Health and Medical Research Council (HREC Reference number: 1281/17) Project title: Working with Aboriginal communities to improve oral health. An evaluation protocol for oral health services provided by the Poche Centre for Indigenous Health in partnership with Aboriginal Community Controlled Health Services, including the Dalang Project. The amendment to this project to cover the National Fluoride varnish workshop was submitted on 16 October 2018 was approved/ratified by the AH&MRC HREC on 6 November 2018.

## 3. Results

Forty-four people attended the Workshop with oral health representatives from all Australian state and territory health departments, as well as the ADA at both the NSW branch and Federal levels. Other representatives came from the Australian Institute of Health and Welfare (AIHW), Australian Human Rights Commission, TAFE, Sydney Policy Lab, Colgate Palmolive, Titanium Solutions, NSW Local Health Districts (Sydney, Murrumbidgee, Western Sydney, Southern NSW and Illawarra Shoalhaven) and the Sydney Children’s Hospital Network.

Eight of the 24 respondents to the pre-Workshop survey were involved in the delivery of fluoride varnish programs, while others were involved in related areas such as policy development and training of dental assistants to apply fluoride varnish. The key areas of involvement of workshop attendees in aspects of fluoride varnish programs from the survey are summarised in Table 1 below.

### 3.1. Oral Health Promotion

One of the main presentations at the workshop looked at the strong evidence for the effectiveness of fluoride varnish as an oral health promotion strategy [11,15,16] and the negligible risk of fluorosis and other side effects [17,18], even in areas with water fluoridation. It was agreed following a whole group discussion that a caries risk assessment undertaken by a dental or oral health clinician is not necessary for the 5–12 years age group if the children are in a high-risk group [20].

Most Australian States and Territories have or are developing fluoride varnish programs for communities with a high population of Aboriginal and Torres Strait Islander people. Tasmania has targeted its state-wide fluoride varnish and sealant program at schools with an Index of Community Socio-educational Advantage (ICSEA) [21] value of less than 1000. These targeting criteria, combined with a high Aboriginal student enrolment, were discussed at the workshop, and were seen as useful criteria for future scale-up efforts nationally.

### 3.2. Systems Alignment and Integration

Several jurisdictions also noted various challenges with implementation and maintenance of fluoride varnish programs including workforce recruitment and travel distances (Table 2). A specific “Apply Fluoride Varnish” unit of competency has been developed for national use in Australia by Registered Training Organisations such as Technical and Further Education (TAFE) Colleges [22] and has been used with dental assistants and Aboriginal health workers in multiple jurisdictions in Australia.

There was strong support for further scale-up of fluoride varnish programs nationally and this was summarised in three key consensus recommendations from the Workshop (Table 3). Each recommendation was formally presented and voted on at the conclusion of the Workshop and endorsed by the majority of those present by a show of hands.

## 4. Discussion

The key outcomes and recommendations of the Workshop were presented at the National Dental Directors meeting on the 16th of November 2018. The national scale-up of fluoride varnish programs and a consistent approach to implementation was endorsed by this group. At a subsequent meeting on the 7th of November 2019, a further update on the Workshop actions was given and, in particular, the use of dental assistants and Aboriginal health workers to apply fluoride varnish and also the use of the Child Dental Benefits Schedule (CDBS) as a funding source were discussed. The CDBS is a national means-tested dental program for children that offers up to $1000 in dental care from private or public dental clinics participating in the scheme [23].

There was strong support at the Workshop related to a standardised protocol, data collection and reporting framework and targeting of schools and preschools. A standard protocol has been piloted and evaluated in NSW and other protocols developed in other States and Territories with the majority targeting primary schools in low socio-economic areas and/or with high Aboriginal populations.

One of the key outcomes of the Workshop was to meet with the ADA about their use of fluorides policy and to seek support for dental assistants to be able to apply fluoride varnish. As a result, the policy was amended in August 2019 to add a new Section 4.2 to the policy, which states that “Topical application of fluorides may also be conducted by appropriately trained Aboriginal and Torres Strait Islander health practitioners or suitably trained dental assistants in remote and very remote regions and in lower socio-economic regions where there is a confirmed need for fluoride varnish application” [4]. It was critical to get their support, in addition to the National Dental Directors, to ensure that the next steps towards a national scale-up, and the greater use of non-dental professionals to apply fluoride varnish, would be supported.

The critical second recommendation was the “Development of an AHMAC Submission seeking support for the National Fluoride Varnish Reporting Framework and also standardized legislation regarding non-registered dental/medical personnel applying fluoride varnish”. Two authors (B.R. and J.S.) met with the NSW Secretary of Health and gained “in principle” support from the Secretary for an AHMAC submission to standardise the use of dental assistants to apply fluoride varnish nationally. This AHMAC submission would include the need for changes to the State and Territory Poisons Acts to allow this to occur, which is also a key strategy of the National Oral Health Plan [1]. A follow-up National Fluoride Varnish Webinar was held on 13 October 2020 to hear from pilot sites in different Jurisdictions, and how further national scale-up can be achieved.

The final recommendation from the Workshop has been largely achieved with broad standardisation on the training requirements for non-dental application of fluoride varnish around the TAFE skillset [22]. This training is being used for both dental assistants and Aboriginal Health Workers. The greater use of Aboriginal dental assistants and Aboriginal health workers may increase participation in the scale-up of fluoride varnish programs, but there is no published evidence of this yet. Nevertheless, a small pilot, and then a subsequent expanded school fluoride varnish program (both in NSW), both found substantially improved participation rates when Aboriginal Education Officers assisted with the consent process [24,25]. At least three Australian states have used the exemptions within their Acts governing the application of fluoride varnish by non-dental professionals. These exemptions do not support the implementation of fluoride varnish programs at scale at a state or national level, as they create a bureaucratic burden to the seek approvals of dental assistants and gain exemptions on a case-by-case basis. A recent scoping review of delivery and financing oral health care noted the importance of considering alternative workforce models [26].

The Workshop also discussed frequency of application of fluoride varnish, with some programs aiming for twice a year while the NSW program aims at four times a year (Table 4). The optimal frequency is between two and four applications per year. The variation in frequency is often based on program logistics including visits by staff to various locations. For example, in the NT, it is linked to biannual child health checks [27], whereas the NSW pilots are based on four school terms per year [24,25]. The NT noted that 6/12 application may limit effectiveness (Table 2). Given the high caries risk of the children and the probability that they may miss one visit in a year, the higher number of applications is preferable. A recent study of Kosovar children found that four applications per year was effective in preventing dental caries in the primary dentition [28], and guidelines in Australia and the United States support this [4,10,16].

Mallee District Aboriginal Services—Mildura, Swan Hill and KerangMurray Valley Aboriginal Cooperative—RobinvaleNjernda Aboriginal Cooperation—Echuca.

It was noted during the Workshop that many fluoride varnish programs are often small scale and episodic due to funding constraints of time-limited grants. One of the authors (J.S.) made a submission to the Fourth Review of the Dental Benefits Act seeking support under the CDBS for the scale-up of school-based fluoride varnish programs and claiming of the relevant items by dental assistants [29]. A recent study of the cost effectiveness of professionally applied fluoride varnish use in Australia found that it was highly cost effective and yielded substantial improvements in oral health outcome and quality of life measures [30]. The authors also recommended that there should be greater use of fluoride varnish via the CDBS [29].

The CDBS provides an appropriate national funding source for scaled-up fluoride varnish programs. The greater use of the fluoride varnish items for programs targeting Aboriginal and Torres Strait Islander may also increase utilisation of the CDBS, which has been particularly low for Aboriginal and Torres Strait Islander children. If dental assistants were also able to claim for fluoride varnish services under the CDBS, this would further improve cost effectiveness while also reducing pressure on oral health services where there are shortages of dental and oral health practitioners [31].

## 5. Conclusions

This study has summarised the promising signs of scale-up of fluoride varnish programs in Australia since the workshop. The Tasmanian program has expanded and three pilot programs in Victoria were undertaken. In NSW, there has been further expansion into urban primary school sites in Sydney. Schools and early childhood centres are feasible settings for the application of fluoride varnish on the teeth of children at high risk of dental caries. There is a need for national leadership to address issues related to the scale-up of the non-dental workforce applying funding for fluoride varnish at scale and their ability to access funding via the CDBS.

## Figures and Tables

**Table 1 healthcare-09-01029-t001:** Fluoride varnish program involvement by workshop participants.

Project Title	Project Overview, Target Population and Setting	Organisation
Northern Territory Healthy Smiles Program	Provide training to a primary health care workforce, for oral health screening, prevention and early intervention for children in primary health care settings of the Northern Territory.	Northern Territory Health
Shaping policy to end tooth decay in Aboriginal children.	Developed and conducted a Fluoride Varnish Skillset training program for Aboriginal dental assistants to apply fluoride varnish in NSW and coordinated delivery of fluoride varnish by Dental Assistants in schools.	Poche Centre for Indigenous Health
Smile Stronger Smile Longer	Fluoride Varnish Program for At Risk Preschool and Childcare Centres within Sydney Local Health District (SLHD). Aims to prevent early childhood caries.	Sydney LHD
Fluoride Varnish Pilot Program for Aboriginal Children in Loddon Mallee Region, Victoria	Aims to prevent tooth decay in Aboriginal children in the Loddon Mallee region, thereby reducing dental pain, adverse impacts on diet and Potentially Preventable Dental Hospitalisations.	Loddon Mallee Aboriginal Reference Group
Oral health and dental care statistics	National Oral health and dental care statistics and reports from the Australian Institute of Health and Welfare (AIHW)	AIHW
Community Fluoride Strategies	A guideline for NSW Public Oral Health Services to work collaboratively with key partners to implement community fluoride strategies to improve the oral health of the population in NSW, particularly groups at high risk of dental caries.	NSW Ministry of Health
ToothSmart Program, Illawarra Shoalhaven Local Health District (LHD)	Reducing early childhood caries in siblings of children on general anaesthetic waiting lists in Illawarra Shoalhaven. Children under 5 years of age, primary school aged children, hospital clinic setting.	Illawarra Shoalhaven LHD

**Table 2 healthcare-09-01029-t002:** Challenges to scale-up of fluoride varnish programs by State and Territory.

State/Territory	Challenges to Scale-Up of Fluoride Varnish Programs
New South Wales	Individual exemptions for dental assistants required to allow them to apply fluoride varnish.
Victoria	Under the current Dental Weighted Activity Unit (DWAU) funding model in Victoria as the cost for Community Dental Clinics (CDCs) to apply fluoride varnish in an outreach clinical setting, within the pilot, is high (approx. $250/child). Few CDCs are providing fluoride varnish applications as a part of their standard outreach programs that may mean duplication of services when provided by the pilot. Low personnel capacity within Aboriginal Community Controlled Health Services to support implementation.
South Australia	Staff and travel costs associated with remote locations; High rate of staff turnover in the various sites; High number of children who were only seen once and missed follow up visits.
Western Australia	Issues with scale up and sustainability given workers have so many other responsibilities; State-wide scale-up in WA is fragmented.
Northern Territory	Staff turn-over; Cost of re-training staff; Travel distances; Only two applications a year may limit effectiveness.

**Table 3 healthcare-09-01029-t003:** Key consensus recommendations from the National Fluoride Varnish Consensus Workshop 2018.

Workshop Recommendations
1. Support for the development of a National Fluoride Varnish Reporting Framework with a standardised protocol, training and targeting of schools/pre-school (in the first instance);
2. Development of an Australian Health Ministers Advisory Council (AHMAC) submission seeking support for the National Fluoride Varnish Reporting Framework and standardised legislation regarding non-registered dental/medical personnel applying fluoride varnish;
3. Certificate III in Dental Assisting with Apply Fluoride Varnish module is the supported level of training with flexibility to include other positions such as Aboriginal Health Workers.

**Table 4 healthcare-09-01029-t004:** Fluoride varnish programs by Australian State and Territory.

State/Territory	Locations	Age Group/Target Group	Setting	Varnish Application Frequency	Workforce
New South Wales	13 rural, regional and urban primary schools	Aboriginal children 5–11 years of age (but some non-Aboriginal children included per community preference)	Schools	313/12	Aboriginal dental assistants; oral health therapists.
	1 rural and 1 urban primary school (SunSmiles Program)	Primary aged children	Schools	6/12	Dental Hygienists/oral health therapists
	Illawarra Shoalhaven, Southern and Murrumbidgee Local Health Districts (LHDs)	Primary aged children (in several schools in each LHD)	Schools	3/12 and 6/12	Dental Hygienists/oral health therapists
	Sydney LHD	Preschool Children	Preschools	6/12	Dental Hygienists/oral health therapists
Victoria	Macedon District pilot (DHSH)	Aboriginal children 4 years of age	Early childhood services	6/12	Oral health therapists.
	Loddon Mallee pilot #	3–18 years of age	Early childhood services and schools	6/12	La Trobe University dental students to apply FV under supervision. Dental therapists and Dentists from Community Dental Clinics
	Latrobe valley pilot	6–7 years of age	Early childhood services and schools	6/12	Dental therapists and Dentists from Community Dental Clinic
	1 rural primary school (SunSmiles Program)	Primary aged children	Schools	6/12	Dental hygienists/oral health therapists
Western Australia	Regional and remote areas of WA since 2015	Aboriginal Children 18 months to 5 years of age		6/12	Aboriginal Health Workers
Northern Territory	The Healthy Smiles Program. Program development commenced in 2009 following the Menzies School of Health Research “Strong Teeth for Little Kids” (STLK)	Aboriginal Children 18 months to 5 years of age	Oral health screenings and application of fluoride varnish have been incorporated into the NT Healthy Under 5 Kids Partnering Families (HU5K-PF) program.	6/12	Aboriginal and Torres Strait Islander Health Practitioners; Nurses and Medical Practitioners. Under the Medicines, Poisons and Therapeutic Goods Act 2012 (NT), Aboriginal and Torres-Strait Islander Health Practitioners, Nurses and other health practitioners use scheduled substances in accordance with Scheduled Substance Treatment Protocol (SSTP), Declared Places and the Gazette notice approved by the Chief Health Officer.
Tasmania	As of September 2019, 82 primary schools were receiving the program	All children 4–12 years (in school with ICSEA index <1000)	Schools	Fluoride varnish 6/12 and fissure sealants on all molars.	Dental and oral health therapists
Queensland	Some LHDs have targeted local schools		Schools	6/12	Dental and oral health therapists
South Australia and Australian Capital Territory	No current preventive structured fluoride varnish programs currently in operation				

Notes: # The Loddon Mallee Aboriginal Reference Group is a forum comprised of Aboriginal Controlled Community Organisations in north-west Victoria and its current membership is Bendigo and Districts Aboriginal Cooperative—Bendigo.

## Data Availability

Not applicable.

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
