# Peer review of "Child Fluoride Varnish Programs Implementation: A Consensus Workshop and Actions to Increase Scale-Up in Australia"

_healthcare, 2021, doi:10.3390/healthcare9081029_

Round 1

Reviewer 1 Report

The article is well written and covers an important topic that could be interesting for the readers.

Fluoride programs are very important in National Health program and they should be applied in every country.

I suggest minor revisions in order to improve some aspects of the article that could improve its value.

Authors could cite a recent article about oral health epidemiology:

Stormon N, Clifford S, Lange K, Mangoyana C, Ford P, Wake M, Lalloo R. Oral health: Epidemiology and concordance in Australian children and parents. Community Dent Oral Epidemiol. 2021 May 28. doi: 10.1111/cdoe.12662. Epub ahead of print. PMID: 34050542.

Authors could outline other methods of reducing caries through the use of fluoride cyting for example water fluoridation:

Shen A, Bernabé E, Sabbah W. Systematic Review of Intervention Studies Aiming at Reducing Inequality in Dental Caries among Children. Int J Environ Res Public Health. 2021 Feb 1;18(3):1300. doi: 10.3390/ijerph18031300. PMID: 33535581; PMCID: PMC7908536.

Mariño R, Zaror C. Economic evaluations in water-fluoridation: a scoping review. BMC Oral Health. 2020 Apr 16;20(1):115. doi: 10.1186/s12903-020-01100-y. PMID: 32299417; PMCID: PMC7164347.

Authors could outline that unfortunatelly literature is full of articles related to bio-active materials or techniques aimed at reducing biofilm formation ant that they are all related to clinical situations after the treatment of caries lesions.

To support the above sentence you could use the following references:

Ionescu AC, Cazzaniga G, Ottobelli M, Ferracane JL, Paolone G, Brambilla E. In vitro biofilm formation on resin-based composites cured under different surface conditions. J Dent. 2018 Oct;77:78-86. doi: 10.1016/j.jdent.2018.07.012. Epub 2018 Jul 17. PMID: 30030124.

Cazzaniga G, Ottobelli M, Ionescu AC, Paolone G, Gherlone E, Ferracane JL, Brambilla E. In vitro biofilm formation on resin-based composites after different finishing and polishing procedures. J Dent. 2017 Dec;67:43-52. doi: 10.1016/j.jdent.2017.07.012. Epub 2017 Jul 24. PMID: 28750776.

Paolone G, Moratti E, Goracci C, Gherlone E, Vichi A. Effect of Finishing Systems on Surface Roughness and Gloss of Full-Body Bulk-Fill Resin Composites. Materials (Basel). 2020 Dec 11;13(24):5657. doi: 10.3390/ma13245657. PMID: 33322405; PMCID: PMC7763061.

On the other hand the authors shall outline that prevention could avoid the above-mentioned consequences outlining the importance and the benefits of a varnish campain in the population like:

  1. reducing overall incidence of caries
  2. reducing costs for caries treatments (private ones or national health services)

You could support the above sentences with the following articles:

Oishi MM, Childs CA, Gluch JI, Marchini L. Delivery and financing of oral health care in long-term services and supports: A scoping review. J Am Dent Assoc. 2021 Mar;152(3):215-223.e2. doi: 10.1016/j.adaj.2020.12.004. PMID: 33632411.

Among the benefits, authors could also outline the role of fluoride varnish in MIH clinical situations (Kumar A, Goyal A, Gauba K, Kapur A, Singh SK, Mehta SK. An evaluation of remineralised MIH using CPP-ACP and fluoride varnish: An in-situ and in-vitro study. Eur Arch Paediatr Dent. 2021 May 31. doi: 10.1007/s40368-021-00630-5. Epub ahead of print. PMID: 34057698.).

Author Response

Authors could cite a recent article about oral health epidemiology:

Stormon N, Clifford S, Lange K, Mangoyana C, Ford P, Wake M, Lalloo R. Oral health: Epidemiology and concordance in Australian children and parents. Community Dent Oral Epidemiol. 2021 May 28. doi: 10.1111/cdoe.12662. Epub ahead of print. PMID: 34050542.

Response

A new epidemiology paragraph has been added to the manuscript with relevant references (see Lines 56-68).

Authors could outline other methods of reducing caries through the use of fluoride cyting for example water fluoridation:

Shen A, Bernabé E, Sabbah W. Systematic Review of Intervention Studies Aiming at Reducing Inequality in Dental Caries among Children. Int J Environ Res Public Health. 2021 Feb 1;18(3):1300. doi: 10.3390/ijerph18031300. PMID: 33535581; PMCID: PMC7908536.

Mariño R, Zaror C. Economic evaluations in water-fluoridation: a scoping review. BMC Oral Health. 2020 Apr 16;20(1):115. doi: 10.1186/s12903-020-01100-y. PMID: 32299417; PMCID: PMC7164347.

Response

Two new sentences have been added to the manuscript with relevant references, including the suggested reference from Mariño et al. (see Lines 48-51).

Authors could outline that unfortunatelly literature is full of articles related to bio-active materials or techniques aimed at reducing biofilm formation ant that they are all related to clinical situations after the treatment of caries lesions.

To support the above sentence you could use the following references:

Ionescu AC, Cazzaniga G, Ottobelli M, Ferracane JL, Paolone G, Brambilla E. In vitro biofilm formation on resin-based composites cured under different surface conditions. J Dent. 2018 Oct;77:78-86. doi: 10.1016/j.jdent.2018.07.012. Epub 2018 Jul 17. PMID: 30030124.

Cazzaniga G, Ottobelli M, Ionescu AC, Paolone G, Gherlone E, Ferracane JL, Brambilla E. In vitro biofilm formation on resin-based composites after different finishing and polishing procedures. J Dent. 2017 Dec;67:43-52. doi: 10.1016/j.jdent.2017.07.012. Epub 2017 Jul 24. PMID: 28750776.

Paolone G, Moratti E, Goracci C, Gherlone E, Vichi A. Effect of Finishing Systems on Surface Roughness and Gloss of Full-Body Bulk-Fill Resin Composites. Materials (Basel). 2020 Dec 11;13(24):5657. doi: 10.3390/ma13245657. PMID: 33322405; PMCID: PMC7763061.

Response

We addressed the suggested second option (see the next response).

On the other hand the authors shall outline that prevention could avoid the above-mentioned consequences outlining the importance and the benefits of a varnish campaign in the population like:

  1. reducing overall incidence of caries
  2. reducing costs for caries treatments (private ones or national health services)

You could support the above sentences with the following articles:

Oishi MM, Childs CA, Gluch JI, Marchini L. Delivery and financing of oral health care in long-term services and supports: A scoping review. J Am Dent Assoc. 2021 Mar;152(3):215-223.e2. doi: 10.1016/j.adaj.2020.12.004. PMID: 33632411.

Among the benefits, authors could also outline the role of fluoride varnish in MIH clinical situations (Kumar A, Goyal A, Gauba K, Kapur A, Singh SK, Mehta SK. An evaluation of remineralised MIH using CPP-ACP and fluoride varnish: An in-situ and in-vitro study. Eur Arch Paediatr Dent. 2021 May 31. doi: 10.1007/s40368-021-00630-5. Epub ahead of print. PMID: 34057698.).

Response

We have clarified the non-clinical focus of our use of fluoride varnish (see Lines 76-78).and have also used the suggested reference Oishi et al to strengthen our argument about an alternative workforce model (see Lines 262-264).

Reviewer 2 Report

This is an interesting report which presents the findings of the National Fluoride Varnish Workshop and subsequent actions to scale-up the use of fluoride varnish in Australia.

Introduction: I think that readers would like to learn more about oral health/dental caries experience in Australian population of children, especially from underprivileged communities. How does dental care for children in Australia look like? What is the level of fluoride in the drinking water in Australia?

Is Duraphat the only varnish used in Australia or just the preparation used in this study?

Authors cite The National Oral Health Plan “legislation covering the use of fluoride varnish and other products varies between jurisdictions, impacting on the ability to implement consistent programs that maximise access to an effective intervention. Relevant legislation and regulations including Poisons Acts, Health Acts and Radiation Safety Acts should be reviewed to remove unnecessary barriers to the provision of dental care”. What unnecessary barriers are included in these Acts?

Discussion: I do not understand the sentence: ”A follow- up National Fluoride Varnish Webinar will be held on 13 October 2020 to further progress the development of an AHMAC paper as well as hear from pilot sites in different Jurisdictions, and the Aboriginal dental assistants involved, as to how further scale-up  can be achieved.” – Was this Webinar already organized?

Conclusions: “There is a need for national leadership to address issues related to the scale-up of the non-dental workforce applying funding for fluoride varnish at scale and their ability to access funding via the CDBS.” – The authors mention the Child Dental Benefits Schedule (CDBS) several times. They should add more detailed information about this founding source.

Author Response

and subsequent actions to scale-up the use of fluoride varnish in Australia.

Introduction: I think that readers would like to learn more about oral health/dental caries experience in Australian population of children, especially from underprivileged communities. How does dental care for children in Australia look like?

Response

A new epidemiology paragraph has been added to the manuscript with relevant references (see Lines 56-68).

What is the level of fluoride in the drinking water in Australia?

Response

Two new sentences have been added to the manuscript with relevant references (see Lines 48-51).

Is Duraphat the only varnish used in Australia or just the preparation used in this study?

Response

We have clarified this with a new sentence in the manuscript with relevant reference (see Lines 76-78).

Authors cite The National Oral Health Plan “legislation covering the use of fluoride varnish and other products varies between jurisdictions, impacting on the ability to implement consistent programs that maximise access to an effective intervention. Relevant legislation and regulations including Poisons Acts, Health Acts and Radiation Safety Acts should be reviewed to remove unnecessary barriers to the provision of dental care”. What unnecessary barriers are included in these Acts?

Response

We have clarified this with a new sentence in the manuscript with relevant reference (see Lines 101-104).

Discussion: I do not understand the sentence: ”A follow- up National Fluoride Varnish Webinar will be held on 13 October 2020 to further progress the development of an AHMAC paper as well as hear from pilot sites in different Jurisdictions, and the Aboriginal dental assistants involved, as to how further scale-up  can be achieved.” – Was this Webinar already organized?

Response

This Webinar has been held and the relevant section of the manuscript updated (see Lines 246-248).

Conclusions: “There is a need for national leadership to address issues related to the scale-up of the non-dental workforce applying funding for fluoride varnish at scale and their ability to access funding via the CDBS.” – The authors mention the Child Dental Benefits Schedule (CDBS) several times. They should add more detailed information about this founding source.

Response

More information about the CDBS has been added (see Lines 220-222).